# Experimental Models of Glaucoma: A Powerful Translational Tool for the Future Development of New Therapies for Glaucoma in Humans—A Review of the Literature

**DOI:** 10.3390/medicina55060280

**Published:** 2019-06-17

**Authors:** Karine Evangelho, Claudio A. Mastronardi, Alejandra de-la-Torre

**Affiliations:** 1Doctorado en Ciencias Biomédicas y Biológicas, Facultad de Ciencias Naturales y Matemáticas, Universidad del Rosario, Bogotá,11121, Colombia; evangelho.karine@gmail.com; 2Neuroscience Research Group (NeurUROS), Escuela de Medicina y Ciencias de la Salud, Universidad del Rosario, Bogotá, 11121, Colombia; mastronardic@hotmail.com

**Keywords:** glaucoma, experimental models, therapy, animals, aging, translational ophthalmology

## Abstract

Glaucoma is a common complex disease that leads to irreversible blindness worldwide. Even though preclinical studies showed that lowering intraocular pressure (IOP) could prevent retinal ganglion cells loss, clinical evidence suggests that lessening IOP does not prevent glaucoma progression in all patients. Glaucoma is also becoming more prevalent in the elderly population, showing that age is a recognized major risk factor. Indeed, recent findings suggest that age-related tissue alterations contribute to the development of glaucoma and have encouraged exploration for new treatment approaches. In this review, we provide information on the most frequently used experimental models of glaucoma and describe their advantages and limitations. Additionally, we describe diverse animal models of glaucoma that can be potentially used in translational medicine and aid an efficient shift to the clinic. Experimental animal models have helped to understand the mechanisms of formation and evacuation of aqueous humor, and the maintenance of homeostasis of intra-ocular pressure. However, the transfer of pre-clinical results obtained from animal studies into clinical trials may be difficult since the type of study does not only depend on the type of therapy to be performed, but also on a series of factors observed both in the experimental period and the period of transfer to clinical application. Conclusions: Knowing the exact characteristics of each glaucoma experimental model could help to diminish inconveniences related to the process of the translation of results into clinical application in humans.

## 1. Introduction

Glaucoma is a leading cause of irreversible blindness worldwide [1]. The data suggests an increase in the incidence of this pathological condition by 2020, with this disease affecting more than 80 million people, 8.4 million of whom are expected to develop blindness [2]. At present, visual impairment and blindness represent a major burden on healthcare systems. For instance, it has been estimated that major visual disorders among individuals aged 40 years or older cost the US $35.4 billion a year [3].

Glaucoma is a neuropathy characterized by a progressive loss of retinal ganglion cells (RGC), their axons, and a concomitant loss of the visual field. Therefore, most pre-clinical glaucoma models are designed to increase intraocular pressure (IOP) to levels that produce a deterioration of the RGC without causing damage to the rest of the intraocular structures [4]. However, some patients develop normal tension glaucoma, which is not associated with an increase in IOP [5], demonstrating that several mechanisms independent of pressure are responsible for the development and progression of glaucomatous neuropathy [6]. When patients are diagnosed with glaucoma, they already have pre-existing ocular damage; this makes it difficult to study the development of the disease from its starting point only with the diagnosis in vivo. This is one of the reasons why animal models are used to study the pathophysiology of glaucoma [6]. However, given this complexity, a single animal model will never precisely model all aspects of all the different types of human glaucoma [7].

Glaucoma can be classified as primary and secondary and in relation to the angle (closed or open) [7] and experimental models have already been developed for both forms of presentation. The most common form is primary open angle glaucoma (POAG), yet the prevalence of distinct types of glaucoma differs for each particular region of the world [5]. In glaucoma, the level of IOP constitutes a major risk factor. Preclinical studies showed that lowering IOP can often prevent RGC loss, but clinical evidence suggests that lessening IOP does not prevent glaucoma progression in all patients. There are other risk factors—such as old age, family history of glaucoma, race, and use of systemic or topical corticosteroids—that are also consistent with the presence of glaucoma. Despite the existence of these risk factors, the reduction of IOP, usually with ocular hypotensive drops, remains the only proven method to treat the disease. For this reason, studies based on experimental models with animals have helped to understand the mechanisms of formation and evacuation of aqueous humor, as well as the maintenance of the homeostasis of IOP. These studies have contributed to the understanding of glaucoma’s etiology and the therapeutic development [6].

Several types of experimental models of glaucoma have been described, such as cell cultures, postmortem eyes, and animal models. All have advantages and disadvantages. The animal models keep the visual system and the animal’s organism intact; therefore, the physiological responses are more similar to those that occur in patients with glaucoma than those observed in the cultures of cells or organs. However, these experimental models are limited by the lack of cooperation of the animals and the need to anesthetize them to perform the procedures [8], which affects the results and represents a risk for the animals themselves.

Given these challenges, there is a need to develop new preclinical animal models to mimic human glaucoma and to try new therapeutic strategies in a more reliable way. When designing glaucoma studies, the model used should provide the minimum severity and duration of the disease necessary to answer the scientific question under investigation or the hypothesis being tested. In addition, factors, such as sex, age, anesthesia used, surgical procedure, and the environment, also affect the outcome, which implies that changing the environment to improve well-being will have an impact on the outcome. This review provides information regarding the choice of the ideal animal research model for the induction of glaucoma, in order to transfer knowledge to human practice. It offers a summary of current preclinical models, describing their advantages and limitations.

## 2. Glaucoma and Aging

Glaucoma is a product of multiple phenomena, involving genetic and environmental risk factors. Cellular aging occurs by the lifelong accumulation of oxidative damage, leading to possible neuronal apoptosis and to the functional deficits of aging, causing the loss of visual function in older adults [9].

Quantitative and qualitative changes occur in collagen structures and cause alterations in the extracellular matrix, generating a more rigid extracellular environment at the ocular level, according to the individual ages [10]. Increased collagen production can provide stiffness in the aged tissue by making it weak and decreasing the elasticity compared to younger tissue [11], with ocular rigidity increasing with age [12]. One of the problems caused by ocular stiffness is the increase in IOP, caused by the decrease in the drainage flow of the aqueous humor [13]. In glaucoma, the main structures affected by changes in the extracellular matrix are the trabecular meshwork and the head of the optic nerve. These structures present more rigidity and greater fibrotic change due to the extensive remodeling of the extracellular matrix [14]. The amount of insults tolerated by the retina produces a “sliding window” effect in the age of onset of glaucoma, increasing the prevalence of this disease in individuals of 40 years [15]. Interestingly, research reports that increasing age is a major risk factor for glaucoma, although several studies show no significant change in IOP with age [15]. It is already reported in the literature that people who suffer from diseases, such as hypertension, and elevated intraocular pressure associated with the age factor are more at risk of developing POAG [16]. Most of the research done in animal models to produce glaucoma used young animals. This factor can alter the results of the study, hindering the understanding of the disease and its transfer to the clinic, since the risk factor of the disease is directly related to the increase in the age of the patients [14]. There are very few experimental studies in the literature comparing the results of glaucoma induction in middle-aged and older animals, and these tend to report inconsistent results. However, when using animals of different ages, we should have some specific care, such as: Weight loss (caused by teeth that are too large, which affects their ability to eat), environmental acclimatization, and animal management (related to stress), as well as the use of anesthetic agents (older animals are more at risk).

The age-dependent ocular hypertensive response to topical steroids has been observed in experimental model rabbits with good resemblance to clinical observations in humans. Topical application of dexamethasone resulted in an increase of intraocular pressure of young rabbits (5 mmHg or more IOP rise in 76%) but had no effect on the intraocular pressure of old rabbits [17,18]. In young rabbits, it is possible to observe alterations in the outflow of aqueous humor due to the immaturity of the iridocorneal angle, these particularities are also observed in young patients. In addition to this particularities in young rabbits, these animals have a greater capacity to respond to steroids which may be due to the increased permeability of the cornea [19].

Chronic glaucoma models developed in mice can also show age-dependent structural changes, such as the progressive thinning of the outer plexiform layer, the decrease in the amplitude of the a and b waves of the electroretinography (ERG) in adult animals, and progressive degenerative structural changes [20]. The extent to which the retina ages generates less ATP, and the ERG tracing may decrease, as observed in the retina of young mice compared to old mice [16]. It is possible to observe a decrease in some layers of the retina in old mice compared to young mice. Also, it is possible to observe that the crushing of the optic nerve occurs more rapidly in older mice than in young mice [21]. For example, there are reports of RGC degeneration that begins quickly following mechanical optic nerve damage. After a severe crush injury in the mouse, RGC survival has been shown to drop to 47% and 27% at one and two weeks post-injury [4]. These characteristics indicate that age affects both the retinal lesion and recovery [14], helping to select the animal model for study.

## 3. Diagnostic Refinement in Humans

Anatomical and functional changes produced by glaucoma are often irreversible. Therefore, early detection remains an important strategy to prevent vision loss. Consequently, the evaluations of the structure and function of the ON and lamina cribrosa (LC), as well as detection of alterations in RGC, remain the best methods for disease prevention [22]. Currently, the main evaluation tools used and tested in practice are tonometry for the measure of the IOP and the visual field test to evaluate the progression of visual field defects [22]. Nowadays, it is common to use ERG, oscillatory potentials [23], and optical coherence tomography (OCT) as methods of early diagnosis of glaucoma [22]. These diagnostic methods are easy to perform and minimally invasive, making it possible to utilize them in animal models [23].

## 4. Models Used for the Understanding of Pathophysiology of Glaucoma

Over recent decades, studies conducted in animal models have contributed to the understanding of the functionality of various clinical and surgical therapies for glaucoma [24]. A wide range of treatment strategies aiming to aid the preservation of visual function has been tested, and yet, until now, there has been little progress in launching new classes of drugs against glaucoma. This lack of innovation has caused a significant revenue decrease in the glaucoma pharmaceutical market. However, some drugs in the experimental phase are in the process of entering the ophthalmology market [25].

Currently, attempts are being made to increase the productivity of drug development through genetic manipulation strategies in human tissues and cells. The use of stem or progenitor cells for cell replacement is a promising strategy due to the demonstrated ability of these cells to induce tissue reconstruction and functional regeneration [26]. However, the absence of predictive animal models limits the opportunities available to study glaucoma and estimate the success or failure of new therapies during clinical trials [25].

Animal models used in glaucoma research should typify the disease with high precision. Due to this, experimental pre-clinical models have to replicate some of the following lesions: Damage of the RGC axons; alterations in the anterior segment; death of RGC; OH induced alteration; as well as alterations in the head of the ON and in the LC [27], and to do so while being practical and cost-effective [28]. The vast majority of pre-clinical studies in the field of glaucoma have been conducted in rodents. Nevertheless, a variety of models are currently being developed in different animal species, including monkeys, rabbits, mice, pigs, and dogs [29,30], all of which have their advantages and limitations (Table 1). Meanwhile, the use of these animals can present problems of ethical restrictions. These restrictions aim to ensure the animal’s welfare as an ethical assessment of a consequentialist nature, evaluating the costs and benefits specific for each species [31]. Because they are in vivo investigations, researchers are obliged to apply the principle of the three R’s described by Russell and Burch [31], in order to minimize suffering or harm for the animals. The three Rs are: Reduce the number of animals used; replace the living animal with alternative experimental techniques; and refine the techniques to minimize animal suffering. These principles have been adopted through bioethical standards enforced by the ethics committees of different countries around the world [4].

### 4.1. Non-Human Primates

Due to anatomic similarities of the iridocorneal angle, retina, and optic nerve head, non-human primates serve as the animal model of choice for glaucoma research [32]. However, the mere fact of anatomical similarity between species is insufficient for it to become an ideal experimental model, as there are potential complications that can occur during the translation of pre-clinical results to human subjects. The complications may involve inadequacy in the course of the disease produced in animals and humans, and a mismatch between visual loss developed in a clinical trial and the one studied in the preclinical model due to the reduced research time in these animals [24]. In this animal model, it is possible to evaluate RGC, as well as to validate non-invasive tests for the onset and chronicity of glaucoma through in vivo measurements of neural structure and function [33]. Moreover, it also serves as an excellent model of ON excavation [34]. With respect to the anatomical resemblance between primate and human eyes, various similitudes have been described, including the anterior part of the trabecular meshwork (TM), the presence of collagen fibers developed in the LC, and a flat disk, as well as the morphology of the ON and the number of RGC [35]. Other characteristics of this model, that are close to humans, are the long lasting IOP elevation and the physiologic processes of the monkey eye [32]. The main anatomical difference between the human and monkey eye is the large number of septa in the Schlemm’s Canal [28]. Overall, a non-human primate model of glaucoma provides several advantages, such as a greater opportunity for studying central effects of the disease, including responses in the central nervous system (CNS) visual centers [24]. However, the use of these animals presents some disadvantages, such as a low availability of the genomic sequence and inaccessibility of transgenic strains as well as rising ethical concerns, high cost, and the need for special facilities alongside the difficulty of both maintenance and handling [28]. All these factors ultimately lead to these animals being used less frequently [24].

### 4.2. Rabbits

Some of the eye structures of rabbits, such as the ON, LC, and astrocytes [36], as well as the composition of the aqueous humor with amino acids and free peptides are very similar to those of humans [37]. Thus, the rabbit eye serves as an excellent model for the analysis of ocular biochemical composition. On the other hand, these animals have a smaller scleral thickness, greater choroidal flow, and a lower vitreous volume of approximately 1.5 mL less compared to humans, which may hinder the production of some glaucoma models and the evaluation of the therapeutic effect on the retina [38]. Although the size of the rabbit eye is generally smaller than that of a human, the dimensions of the anterior chamber (AC) are generally larger than those of people, facilitating the use of ophthalmological techniques in the anterior segment. The lens of rabbits is usually larger and thicker, occupying more space of the vitreous cavity [39], which could complicate the collection of the posterior segment for biochemical analysis. However, this particularity of the lens can lead to increased safety of a transfer of the experimental technique to human subjects. In albino animals, the pigmented epithelium of the retina does not produce pigmentation, which facilitates the evaluation of possible toxic effects of drugs, since generally ocular pigment plays a protective role against toxicity [36]. It is also observed that the response to ON damage presents both clinical and experimental characteristics similar to those of humans with glaucoma [37]. This animal model represents a simple, low-cost, repeatable, and reproducible alternative for pre-clinical eye disease research [18]. Rabbits are docile, have an appropriate eye size for the use of certain techniques [40], and have a relatively long life span in comparison to other animal species [18].

### 4.3. Rodents

Because of the powerful genetic tools available for use in laboratory mice, they have proven to be a highly useful mammalian system for studying the pathophysiology of human disease. The similarity between human and mouse eyes coupled with the ability to use a combination of advanced cell biological and genetic tools in mice have led to a large increase in the number of studies that use murine models of specific glaucoma phenotypes [27].

Currently, possible causal variants identified in human genetic studies can be validated in mice, allowing reporting on new genes and pathways that impact glaucoma [41] and improving the outcomes of translational studies. Furthermore, over the past few years, experiments in human and mouse cells revealed that some transcription factors may be equally potent in cell reprogramming [42,43]. These experimental findings have made stem cell technology increasingly accessible to laboratories without previous experience with stem cells and are now so optimized that derivation, maintenance, and cell differentiation are a widely used tool for biomedical research [43]. Increasing numbers of studies on stem cells are being carried out due to their curative potential and prospects for becoming an alternative to conventional drugs [44]. With the development of CRISPR-Cas9 technology, any limitations to the full potential of genome editing are being eliminated. Hence, it is becoming possible to edit altered genes in pluripotent stem cells of patients with certain diseases to neutralize the mutation [43,45]. This technology is being used in mice, and its application for glaucoma research promises to be informative [45].

There is extensive data on the biology of the central nervous system of rodents and how damage at the retinal and ON level is related to increases in IOP. This animal model allows for the study of the effects of the ON lesion independently of ischemia in the RGC population, owing to the presence of the central artery of the retina that can be cut without compromising retinal circulation in rodents [7]. The ocular anterior segment of rodents is examined in studies dealing with relevant aspects of the physiology of glaucoma, since these animals present similar regulatory mechanisms of the production and flow of aqueous humor to humans, and respond alike to IOP-reducing drugs [46]. Despite being presented as an optimal animal model for the induction of glaucoma, the size of the eye and the handling of these animals can deter the use of tonometry and some surgical techniques [36]. Likewise, morphological features of the optic nerve head (ONH) and TM can limit the use of these animals in the OH model. The ONH does not have collagen fiber and instead contains pores created by astrocytes [47], while the TM may not generate high flow resistance since it only has a few layers of cells [28]. The short life span of these animals is another issue that restricts the demonstration of disease progression [24] and hence compromises the translation of research data to clinical trials in patients.

The use of both rats and mice has the same advantages: They both have a general similarity with the human eye, a similarity in the dynamics of aqueous humor, experimental elevation of the IOP, and excavation of the ON. While mice have better responses in comparison to rats in relation to the spontaneous increase of IOP, the availability of the genomic sequence, and transgenic strains, it is more problematic to develop tests for mice than for rats [8]. As mice lack a collagenous LC, this animal is not recommended for research on specific functions of this structure. In rats, the RGC layer hosts a small population of displaced amacrine and horizontal cells [48], which means the anatomical criteria are based on the size of the neuronal soma, thus providing less reliable results for the transfer of research data to the clinic [7].

Despite these disadvantages, rodents remain the most commonly used animal model for research on glaucoma. Among the reasons why, is the high degree of conservation between human and rodent genomes, the availability of an established technique for genetic manipulation, and the convenience of breeding. Due to the low cost and relative ease of obtaining eyes, experimental samples in this model can be large [37].

### 4.4. Glaucoma Models in Other Animals

Nowadays, unconventional animals, such as pigs, mini-pigs, and zebrafish, are being used for modelling glaucomatous disease, as these animals are more accessible and their use in research raises fewer ethical concerns [29].

The pig’s and mini-pig’s retina have more similarities with the human [49] than the retina of any other mammal, such as the dog, cat, goat, and cow [29]. The mini-pig model of glaucoma presents a typical alteration in the ON, with a predominant involvement of arterioles serving as a study tool for this structure. However, this model offers a challenging visualization of the LC due to the central venous ring location in the center of the optic disk [50]. Regarding the ocular anatomy of pigs, their retina appears to be very similar to humans with the presence of holangiotic retinal vasculature, without tapetum, with cone photoreceptors located in the external retina, a similar scleral thickness [51], and the presence of three types of RGC. These characteristics help to elucidate the mechanisms involved in the selective death of RGC that are believed to be vulnerable in clinical and experimental glaucoma [52]. In addition, this model is capable of producing IOP elevation and is crucial for research on ocular regeneration using stem cells [53]. These animals are considered easy to handle, grow slowly, and have a suitable eye size for the use of diagnostic tools, such as OCT, corneal topography imaging, or ERG [29,54].

Although there are some differences in the ocular anatomy between zebrafish and humans, it is possible to reproduce chronic IOP elevation, loss of RGC, and progressive damage in the ON in these animals [55]. The eye anatomy of zebrafish has some similarities to the human ocular structure. Similitudes include the retina, ON as well as the development of a superficial ectoderm, neuroectoderm, and mesenchyme of the head; however, the lens is rounder than in humans [56]. Zebrafish behavior is an invaluable tool for analyzing visual function. Zebrafish alter their skin pigmentation when exposed to different light intensities by expanding or contracting melanosomes; if a fish has impaired vision, it perceives itself to be in an environment with low light intensity, therefore appearing hyper-pigmented [55]. Insertional mutagenesis is a powerful tool for the search and determination of the function of genes involved in the development of vertebrate organisms [57] and in disease progression, for instance, in glaucoma [58]. The mapping and characterization of many genes important in maintaining normal eye development in the zebrafish have also been identified through reverse genetic or genotype-based analysis [59]. Germline transgenic fish can be generated in order to evaluate the role of the over-expression and misexpression of candidate disease genes. However, the application of antisense modified oligonucleotides, known as morpholinos, is most often used to inhibit specific gene functions [60]. For example, while there is just one *PAX2* gene in humans, there are two coding genes in zebrafish: *Pax2.1* and *pax2.2*. Careful study of duplicated genes has led to the general observation that most often, the two duplicated genes diverge over time within regulatory sequences [58]. The hypothesize that the mutated LMX1B may act in concert with another inherited mutation to promote glaucoma in more subtle ways due to a small elevation in IOP and a secondary mutation may predispose ganglion cells to be more sensitive to elevated IOP [58]. The zebrafish is an ideal model to carry out ambitious projects with moderate budgets [61] due to its short production time, large number of offspring, and easy handling [56]. One of the limitations that makes translational research rather challenging [24] in this animal model is its small size, since the anterior segment of the eye has an average diameter of 1.4 mm and a maximum average depth of 0.25 mm in the peripheral angle, thus it is difficult to use diagnostic methods, such as OCT, tonometry, and ERG [58].

## 5. Diagnostic Refinement in Animals

When choosing an animal model for studies involving particular tests, the researcher should be especially cognizant of the differences in the performance of the procedure between pre-clinical study and applications in humans, as in the case of ERG [62,63]. The use of anesthetics to perform ERG in animals is known to affect neurotransmission, consequently altering test results [64]. Other relevant parameters are body temperature and age. A significant decrease in the animal’s body temperature leads to an overall decline in metabolism, diminishing the intensity of chemical reactions and the amplitude of ERG in rats and rabbits [65]. Age, on the other hand, can also interfere with the test results; for instance, young rabbits of less than 3 months of age exhibit smaller b-wave amplitudes than older rabbits aged 15 to 27 months [66]. Thus, age is an important data consideration when transferring experimental results to clinical studies in humans.

The IOP values vary by animal breed, method of sedation, and measure. In general, excessive restraint, inadequate positioning, or lack of experience in the use of equipment can increase IOP [29]. This information is very important when reviewing preclinical results of drugs that reduce IOP and during translation to clinical trials in humans, since the reduction in IOP may occur due to the employed technique and not due to the studied drug [67]. When measuring IOP in rabbits, one must be extremely cautious while standardizing the results, since stress can provoke eye dryness, thus altering the measurements [68]. Some other variables, such as blood pressure, pulse, respiration, and anxiety, may also affect IOP measurement, modifying the reliability of results [67,69].

## 6. Genetic Models of Glaucoma

The main advantage of working with genetically modified animals compared to surgically induced models is a more uniform response in terms of the elevation of IOP, and damage to the retina and ON. Importantly, the use of modified animals significantly aids the identification of interactions between loci that cause complex disease [59]. However, the time needed for these animals to develop glaucoma is usually too long (9–15 months) and cannot be adapted experimentally [4].

In current practice, animal models with certain genetic characteristics are established as inbred strains, for example, DBA/2 J mice. This strain of mice is used as an experimental model of POAG, since they develop a progressive chronic increase in IOP that causes the death of RGC [69]. The increase in IOP in these animals starts at 8 months of age, and the pressure remains high until death [29].

The use of the inbred animals, such as the same age, sex, weight, and stage of the disease, may decrease experimental variability, which would make it difficult to transfer preclinical data to a clinical trial. These conditions cannot be met in clinical trials as patients vary in age, stages of the disease, compliance with the use of therapy, and visual performance, as well as having different pharmacological profiles [24]. These particular characteristics must be taken into consideration by the researcher when choosing the aforementioned animal model for the experiment, as the results may be significantly different in animals compared to clinical research in human subjects.

Progress in the treatment of glaucoma has been linked to the development of transgenic models. One of these models exhibits a specific mutation in the gene of alpha-1 subunit of type 1 collagen linked to a gradual elevation of IOP and progressive loss of the ON axon [70]. Transgenic mice expressing this form of collagen develop IOP that is elevated by almost 5 mmHg compared to controls at 36 weeks. These mice demonstrated progressive optic nerve axon loss with normal organization of the drainage structures [4]. Mutations affecting serine protease (PRSS56) produce a certain type of phenotype in mice that is similar to angle-closure glaucoma with a reduction of the ocular axial length, a relatively large lens, and a narrow angle [71]. In addition to these models of glaucoma, we highlight a model capable of reproducing the degree of elevation of IOP, degeneration of the peripheral optic nerve, and damage produced in the retina of humans through the formation of transgenic mice expressing the Tyr437His mutation of human myocilin protein with the purpose of producing primary open angle glaucoma [72,73]. To induce normal-tension glaucoma, mice with a deficiency in the glutamate/aspartate transporter (GLAST) are used. These animals show typical glaucomatous damage, such as death of RGC (due to greater sensitivity to oxidative stress) and degeneration of the ON [74]. The incidence of congenital glaucoma in humans is less than 1% and is associated with a number of transcription factors, including: *FOXC1, FOXC2, PITX2, LMX1b*, and *PAX6* [4]. Targeting of NC-Foxc1 and NC-Foxc2 enables the identification of malformations in the embryonic period as the joint expression of these genes regulates early development of some ocular structures [75]. The knockouts of these genes are lethal in the embryonic or neonatal periods. Foxc1 −/− mice die at birth and Foxc1 +/− animals have defects in the ocular drainage structures without changes in the IOP [4]. Conditional knockout mice for Foxc2 are generated by the elimination of Foxc2 from cells derived from the neural crystal. These animals suffer from ocular disease [75] and are used to study the development of the ocular drainage angle. *CYP1B1* (cytochrome P450, family 1, subfamily b, polypeptide 1) is a gene suggested to be involved in congenital glaucoma. *CYP1B1* Knockout (KO) animals are capable of developing ocular abnormalities similar to those observed in humans with primary congenital glaucoma, for example, small or absent Schlemm’s canal, defects in the TM, and fixation of the iris to the TM and peripheral cornea [76].

## 7. Models of Ocular Hypertension (OH)

Different experimental OH models have been employed in order to determine the effects of increased IOP on neuronal populations of the retina and to define the changes occurring in the external retina after the elevation of IOP [77].

Animal models of OH developed to imitate POAG show moderate increases in IOP that produce damage at the level of the ON. For this reason, reaching extremely high pressures in the induction phase of the disease should be avoided as it could alter the progressive nature of the damage [78].

Various techniques are employed to chronically elevate IOP in experimental glaucoma models in the short term and over an extended period of time under controlled conditions [4,36]. These include laser photocoagulation of the TM, injection of hypertonic saline into episcleral veins [79], cauterization of the episcleral veins, and the injection of substances into the AC to obstruct the aqueous outlet [22]. These methods differ in pressure increase, as well as in the level of control the researcher has over these parameters [7]. As such, researchers should carefully consider which models are best for their particular experiments and the resulting data that can be applied in the clinic. It should be noted that in order to obtain an optimal result in research, it is necessary to provide the animals with adequate accommodation conditions and good acclimatization, as well as a balanced diet, the use of aseptic techniques, and control procedures with anesthetic and analgesic protocols that guarantee the welfare of the animals. In addition, the animal should be monitored constantly and humane endpoints should be determined in advance with the veterinary staff.

The laser photocoagulation technique involves damaging TM, blocking the elimination of aqueous humor, and consequently increasing IOP [4]. The monkey model with laser-induced ocular hypertension mimics clinical glaucoma in humans in respect to aqueous humor inflow and trabecular outflow, two important parameters in the control of IOP [80]. This technique requires at least three laser applications with 7-day intervals to increase IOP by 60%. Major disadvantages of this method are the production of anterior peripheral synechiae, hyphema, and corneal edema [81], as well as its high cost [65]. Modulation of the intensity, duration, and number of laser points provides some control over the duration and range of elevation of the IOP [4].

Mittag et al. comments that in order to investigate cellular processes and mechanisms of RGC death in mice with a diode laser, it is necessary to combine the damage to the TM and scleral veins, using the laser setting at a power of 0.4 W and duration of 0.7. However, if the researcher aims to study the neuroprotective effect of a novel drug, it may be beneficial to use the same power and duration setting, while making a direct application in the TM. This way the damage occurs more slowly, facilitating the detection of the effect of proposed therapy [81].

Use of the argon laser coupled with a slit lamp shows a particular feature of pigment affinity. The presence of pigment in the camera angle drastically alters the absorption of laser energy and the extension of damage produced in the TM [82]. Some researchers have used this feature to their advantage by administering pigment particles in the AC or using pigmented rats to increase energy absorption. For example, there are reports of utilized injection of carbon particles into the anterior chamber of Wistar rats to argon laser treatment directed at the TM. The carbon particles, which by the time of the laser treatment had accumulated within the anterior chamber angle, absorbed laser energy and produced focal heat, thereby generating a localized scarring effect [83]. However, characteristics of the latter model make laser absorption high and inconstant, hence exacerbating the inflammation, which in turn results in a great variation in the subsequent elevation of IOP [4]. Therefore, this model can only be used for certain strains of rodents.

The use of laser photocoagulation combined with temporary narrowing of the iridocorneal angle by paracentesis of AC is also described in the literature. This model is capable of causing an extremely abrupt IOP increase for 3 weeks and a significant decrease in the number of RGC that does not occur during POAG in humans. Other disadvantages of this technique are the presence of inflammation or ischemia and the opacity of the central cornea [84].

A chronic increase in IOP can also be induced by means of a model in which 50 μL of hypertonic saline solution is injected into the collecting veins of a rat’s eye, producing sclerosis of TM and ON damage [78]. In a study of 20 consecutive animals, only 9 had an increase in IOP following a single injection, while subsequent injections raised the intraocular pressure in 7 others. In addition, the mean sustained pressure elevations ranged from 7 to 28 mmHg and the retinal vasculature remained perfused in all eyes [78]. For this model of chronic glaucoma, it is necessary to use specialized microneedles extracted from borosilicate glass micropipettes, attached to a 1 mL syringe, to provide sufficient flow pressure at the tip of the needle. It is necessary that fluids employed in this system are not viscous. To perform this technique, a preceding lateral canthotomy and the application of a plastic ring around the eye is necessary to increase the pressure in other aqueous veins and to drive the sclerosing agent to the limbus. The microneedle should be pointing towards the limbus placed in a position parallel to the vessel wall. At the moment of injecting the solution, special attention should be paid to the coloration of the vessel, since it should remain pale [78]. It should be noted that this technique must be applied in anesthetized animals.

The model based on the application of hypertonic saline solution in the episcleral vein results in a moderate elevation of IOP within 7 to 10 days of the application, with a considerable level of variability between 0 and 30 mmHg [78]. Norwegian brown rats are used for this model due to their docile behavior, which permits daily IOP measurements without the need for general anesthesia [24]. When there is no increase in IOP after initial injection, the procedure can be repeated [54,85]. In this model, OH tends to persist for longer periods of time in comparison to laser-induced OH [86]. This model has been shown to produce progressive loss of RGC, degeneration of ON, and specific changes in ERG, making it similar to other forms of OH [85]. It is generally recommended for studies on neuroprotection [24]. The drawback of the model is that it requires highly specialized materials, injections are technically difficult to perform [87], and extensive training is required. This technique is not yet described in mice, probably due to the small size of the eye, which makes injection into the vessel more difficult [79].

The POAG model can be induced in mice and mini-pigs using the episcleral vein cauterization technique [29]. This model consists of exposing the ocular surface by retracting superior and lateral rectus muscles and applying stimulation by cautery at low temperatures directly in the episcleral veins. Causing a blockage of the drainage of aqueous humor through collecting ducts of Schlemm’s canal eventually leads to a significant increase in IOP [88]. In addition to ocular venous congestion [89], an obstruction of the outflow of aqueous humor has also been described in this experimental model. For the elevation of IOP, more than two veins must be cauterized. Data from an animal study with mice indicates that cauterizing four veins brings the IOP value close to 60 mmHg [26,49]. In this model, RGC cell death due to apoptosis in the peripheral region is close to 4% weekly [4].

Another method for vein occlusion is an application of a nylon suture upon muscle retraction and ocular exposure. With this procedure, the moderate increase in pressure that is achieved immediately after surgery may last for several months. The main drawback of this technique is the possibility of blood vessel reformation with a consequent decrease in IOP that brings the need to repeat the procedure [90].

It is known that topical, periocular, and even systemic or inhaled administration of steroids can cause OH [91]. The significant elevation of the IOP can occur in a matter of hours or up to 1 day after steroid use. Drug discontinuation usually leads to normalization of IOP, however, this side effect may not be reversible in certain individuals [92]. The exact mechanism by which steroids stimulate an increase of IOP is yet to be elucidated. Nevertheless, it is known that a steroid causes stabilization in the lysosomal membrane and the accumulation of glycosaminoglycans polymerized in the TM, generating greater resistance to the outflow of aqueous humor. In addition, these drugs also increase TM resistance due to increased expression of fibronectin, elastin, and laminin [93].

Glaucoma induced by corticosteroids mimics human POAG. Unlike most experimental models of glaucoma, glaucoma generated by corticosteroids is also observed in ophthalmological practice and is generally caused iatrogenically [94]. This type of glaucoma has been mostly studied in the New Zealand breed of rabbits [39]. This animal model mimics several characteristics of the aforementioned condition in humans and provides morphological and molecular evidence on the pathogenesis of glaucomatous disease [95]. Moreover, there are many clinical, morphological, and molecular similarities between corticosteroid-induced glaucoma and POAG, which makes it an attractive model to study specific aspects of glaucomatous pathology [29].

The angle of AC plays a key role in regulating the outflow of aqueous humor and IOP. TM presents sheets containing lamellae of extracellular matrix materials, which comprise an important part of this tissue and probably of the exit flow barrier. Among the materials of the extracellular matrix of TM, some, including glycosaminoglycans (GAGs), hyaluronic acid (HA), keratan sulfate, heparan sulfate, and sulfate-chondroitin, have been identified in rabbits [96], monkeys, and humans [97].

An experimental model of POAG has been developed by application of HA in the AC in rats and rabbits. Single application of 25 μL of 1% hyaluronic acid in the AC of rats produces a significant increase in IOP that can last, on average, up to 8 days. When using the same protocol weekly, a sustained increase in IOP was observed [98]. The effect of maintaining high IOP with a single HA injection may be attributable to the fact that the AC in the rat eye is very shallow. Consequently, the intracameral concentration of the drug is higher than that achieved in rabbits [99] and also than the one expected in humans. During the performance of this technique, the needle direction requires special attention [97], since it can come into contact with the iris and cause a hemorrhage. The needle passing through the corneoscleral limbus must be beveled downward to form a greater angle with the AC and to avoid contact with the lens [99].

Rhythmic oscillations of IOP are a common feature in both humans and laboratory animals [98]. In humans, there are clinical studies reporting that the peak IOP occurs in the morning or during the night [100]. However, the IOP may present variations due to the individual’s postural changes. The postural change from upright to recumbent elevates IOP, because of the hydrostatic responses in the episcleral venous pressure and the distribution of body fluid. Thus, the recumbent body position in the nocturnal/sleep period sets IOP at a different level compared with the upright body position in the diurnal/wake period [100]. Studies with intracameral injection models in rats have consistently shown an increase in nocturnal IOP, presumably due to an increase in the production rate of aqueous humor in this period [99]. Even though the mechanism of IOP elevation related to the application of viscoelastic drugs is still not clearly understood [99], it is apparent that the injection of HA in the AC produces an abnormal accumulation of glycosaminoglycans (GAGs) in the TM, complicating the drainage of aqueous humor [101].

Injection of HA in the AC provides several advantages over other experimental models of glaucoma induction: It is cheap and easy to perform, it allows high and long lasting hypertension to be obtained, and it does not prevent the elimination of aqueous humor, which favors the conduction of pharmacological studies [100].

Another model of experimental glaucoma capable of reproducing POAG is performed in rats via the induction of reperfusion ischemia. It consists of performing paracentesis in the AC of the animal eye. IOP elevation is produced by a 30 G gauge needle connected to 0.9% saline solution at a height of 150 cm above the eye. For example, in this method, the cannulation of the anterior chamber of mice and rats with a microneedle allows one to precisely control the IOP at 110 mmHg, and blood flow through the retinal and uveal vasculature is suppressed. IOP can be normalized by reducing the pressure of the perfusion system after the ischemic exposure period is over. While this method involves extreme acute ocular hypertension, the neurodegenerative effect is thought to be mediated primarily through the ischemic insult, though it is possible that other IOP-induced damage to RGC occurs [4]. This procedure is performed for 60 min and allows an IOP of approximate 70 mmHg to be reached [102].

## 8. Angle—Closure Glaucoma and Neovascular Glaucoma

The experimental model for the development of angle-closure glaucoma (ACG) is described in dogs of several breeds, including beagles, cockers, and basset hounds. Beagles express the autosomal recessive phenotype and begin to develop the disease between 6 and 12 months of age. At the age of 2 to 3 years, as glaucoma progresses, the drainage angle begins to close and the excavation of the ON head can be observed [103]. Other animals are used for this glaucoma model as well, for instance, genetically modified mice. These animals have characteristics that predispose them to angle closure and high IOP, including a reduction of the ocular axial length, a relatively large lens, and a narrow angle [29].

Elevation in the levels of nitric oxide in the aqueous humor has been observed in both neovascular (NVG) and ACG types of glaucoma. This is attributed to an inflammatory manifestation of AC [88]. Therefore, substances that are used in research on these types of glaucoma shall not interfere with study outcomes, as there is a known effect of inflammation on IOP. The IOP elevation from magnetic microbeads is an experimental method that is intended to occlude TM through the use of a magnet that is directed towards the iridocorneal angle [104]. For example, the injection of 10μm latex microspheres with or without the addition of hydrox-ypropylmethylcellulose (HPM) also blocked drainage through the trabecular meshwork and increased intraocular pressure [94]. In this case, nine and six weekly repeated injections, respectively, were necessary before a sustained increase in IOP was achieved, which lasted for at least 30 weeks in each case [105]. The main advantages of this method are that the ocular fundus can be easily examined, since the microspheres do not obstruct the pupil, and that the number of injections can vary to modulate the elevation of the IOP. However, one of the limitations of this method is a fluctuating increase in IOP. The location of microspheres within the AC can also be difficult to control, compromising observations of the ON.

## 9. Future Research

In addition to the eye being a site of immune privilege, it also presents the blood-retina barrier, which makes it difficult for some chemical substances to enter. For this reason, neuroprotective approaches have shown great promise in both in vitro and in vivo studies [9]. The treatment can delay progression of glaucomatous optic neuropathy by lowering intraocular pressure medically or surgically [106]. New treatments targeting newly discovered neuroprotective pathways will enter clinical trials in the coming years and patients’ lives will be improved by the remission or correction of disease pathologies and aging [9]. An alternative promising approach is to use stem cells as a neuroprotective effect. This therapy promotes neuroprotection and functional preservation of RGC, making it a good candidate as an adjunctive therapy to IOP-lowering medications, and thus a potential future treatment for glaucoma [107].

One of the biggest challenges of ophthalmological research in the last decade has been finding a method to replace damaged cells by a neurodegenerative process during ocular diseases without creating unnecessary risks for patients. Soon, it will be important to ethically discuss genome editing with stem cell technology. This topic will require the participation of multidisciplinary groups of specialists in order to support scientific progress and allow for rational risk/benefit evaluations of this technology. To progress in this field, the method of implementation should foster progressive accumulation of knowledge, for which more information from studies with animals whose anatomy and physiology are relevant to humans is needed. This would permit the design of clinical trials that, in addition to being safe for both animals and patients, generate interpretable results. The information obtained with tests designed in this way will allow, in turn, rational modifications of protocols to gradually optimize their results.

## 10. Conclusions

Although there is no perfect experimental model, each of the existing ones has been used successfully to unravel important aspects of glaucoma pathology and could be used to develop new therapies for this disease in the future. However, glaucoma models are not exact recapitulations of the human condition. Therefore, the results obtained in pre-clinical studies should be extrapolated only to the extent that a particular model and the studied research question permits. Overall, we conclude that an ideal and reproducible glaucoma model should be easy to induce, cheap, and as similar to the glaucomatous pathology of humans as possible. Regarding the choice of animal species, the size of the animal’s eye and its similarity to humans should be considered.

## Figures and Tables

**Table 1 medicina-55-00280-t001:** Advantages and limitations of different animal models of glaucoma.

Animals	Induction of Glaucoma	Advantages of the Model	Limitations of the Model
Monkey	Laser photocoagulation Intracameral injection of microspheres	Anatomical similarity with the human eye Similarity in AH dynamics Excavation of ON Experimental elevation of IOP	Genomic sequence Transgenic strains availability High cost Simplicity of maintenance and handling Ease to develop experiments
Rat/Mouse	Injection of hypertonic saline rat into episcleral veins Cauterization of episcleral veins Laser photocoagulation Reperfusion ischemia HA Intracameral injection Intracameral injection of microspheres Genetic	Similarity in AH dynamics Excavation of ON Experimental elevation of IOP Simplicity of maintenance and handling Ease to develop experiments Low cost	Spontaneous elevation of IOP Genomic sequence
Rabbit	Topical application corticoids Laser photocoagulation HA Intracameral injection	Ease to develop experiments Anatomical similarity with the human eye Similarity in AH dynamics Spontaneous elevation of IOP Simplicity of maintenance and handling Low cost	Genomic sequence Transgenic strains availability
Pig/Mini pig	Cauterization of episcleral veins Laser photocoagulation	Similarity in AH dynamics Excavation of ON Experimental elevation of IOP Simplicity of maintenance and handling Low cost	Ease to develop experiments Genomic sequence Transgenic strains availability
Zebrafish	Genetic	Genomic sequence Transgenic strains availability Anatomical similarity with the human eye Spontaneous elevation of IOP Low cost	Ease to develop experiments
Dogs	Genetic	Excavation of ON Experimental elevation of IOP	Ease to develop experiments

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
