# Peer review of "Experimental Models of Glaucoma: A Powerful Translational Tool for the Future Development of New Therapies for Glaucoma in Humans—A Review of the Literature"

_medicina, 2019, doi:10.3390/medicina55060280_

Reviewer 1 Report

Overall, this is an interesting topic which needs to be looked at more thoroughly. An overview of the literature and the pros/cons of each model is a useful tool.

This manuscript needs a good deal of refinement in order to be helpful. The author tends to jump from subject to subject without completing the thought. There are frequent run-on sentences which adds to the readers difficulty in understanding the points of the manuscript.

There is a disconnect between the introduction and the body of the manuscript. The introduction is quite long and does not adequately "introduce" the main topic which is examining different models for studying glaucoma.

There are frequent grammatic and spelling errors which also make the thought flow difficult to follow. Instances occur where the word "sclerotic" is used when I think the author intends to say "sclera".

Section 2 is particularly problematic and difficult to follow. The thought process jumps from retinal issues, to glaucoma risk factors, to normal tension glaucoma, to angle development without any segues in between.

Section 3 suggests OCT as a novel approach to evaluating patients when it is in fact the current standard of care for glaucoma and many other ocular diseases.

Section 4 which is likely the most important part of the manuscript seems disjointed. It is very difficult to follow the thought flow and the points when examining each animal model. I am not clear why the use of some animals draws ethical concerns but others such as pigs do not. There is also little discussion of the actual mode of inducing glaucoma in each model. The mode of induction is brought up in table one but then it is not discussed. Given that these models do not have the same cause for glaucoma as humans and are purely pressure induced, it makes the generalizability of the data difficult.

Overall, I think the authors attempted to discuss too many topics for one manuscript. Between animal models, multiple types of glaucoma, types of induction of glaucoma in animal models, genetics, gene therapy, anatomy and pharmakintetics, this manuscripts broad coverage of topics makes it difficult to follow. The authors should consider breaking the topics apart for different manuscripts.

Author Response

-Overall, this is an interesting topic which needs to be looked at more thoroughly. An overview of the literature and the pros/cons of each model is a useful tool.

We appreciate this comment. We have made the suggested change.

-This manuscript needs a good deal of refinement in order to be helpful. The author tends to jump from subject to subject without completing the thought. There are frequent run-on sentences which adds to the readers difficulty in understanding the points of the manuscript.

We have made the proposed change as it also improves the wording and understanding of the text.

- There is a disconnect between the introduction and the body of the manuscript. The introduction is quite long and does not adequately "introduce" the main topic which is examining different models for studying glaucoma.

We fully agree with the reviewer and have made the suggested change. In this part, we changed several aspects of the introduction, from improving the connectivity between the introduction and the body of the manuscript, to making relevant comments regarding the use of different models for the glaucoma studio.

-There are frequent grammatic and spelling errors which also make the thought flow difficult to follow. Instances occur where the word "sclerotic" is used when I think the author intends to say "sclera".

We appreciate this assessment. The manuscript was edited a native English speaker colleague for corrections.

-Section 2 is particularly problematic and difficult to follow. The thought process jumps from retinal issues, to glaucoma risk factors, to normal tension glaucoma, to angle development without any segues in between.

Thanks for your comments. In this section, we emphasized the use of experimental models with appropriate age for research. The parts that did not follow the sequence of the text were removed.

-Section 3 suggests OCT as a novel approach to evaluating patients when it is in fact the current standard of care for glaucoma and many other ocular diseases.

We have redrafted the phrase so that the idea we want to convey is clearly expressed.

- Section 4 which is likely the most important part of the manuscript seems disjointed. It is very difficult to follow the thought flow and the points when examining each animal model. I am not clear why the use of some animals draws ethical concerns but others such as pigs do not. There is also little discussion of the actual mode of inducing glaucoma in each model. The mode of induction is brought up in table one but then it is not discussed. Given that these models do not have the same cause for glaucoma as humans and are purely pressure induced, it makes the generalizability of the data difficult.

We agree with this reviewer's assessment and have made the suggested change to what refers to ethical standards regarding the use of animals, we had lacked more information about it. For this, we add a paragraph clarifying the reason why some animal models can present ethical problems. Regarding the table, we believe that it contains the relevant information for people who plan to use animals as an experimental model, if we describe the same information in the text the document’s reading could become difficult, so we have maintained the initial idea.

We agree that the causes of glaucoma in humans are not the same in the models of experimental animals because of this, throughout the manuscript we include comments regarding the characterization of a model "that approaches" the disease, since it is still no ideal glaucoma model.

-Overall, I think the authors attempted to discuss too many topics for one manuscript. Between animal models, multiple types of glaucoma, types of induction of glaucoma in animal models, genetics, gene therapy, anatomy and pharmakintetics, this manuscripts broad coverage of topics makes it difficult to follow. The authors should consider breaking the topics apart for different manuscripts.

We appreciate your contributions in this manuscript. We would like to emphasize that in carrying out this review we tried to systematize as much information as possible about the experimental models of glaucoma that have developed in different types of glaucoma, identifying their individual strengths and weaknesses. We know that when deciding which is the best experimental model to use in research, in order to reproduce the mechanisms of glaucomatous disease, the researcher must make a series of previous considerations about the experimental method and the tools to achieve the desired objectives. These considerations, as well as some details that determine the obtention of optimal result are not easy to find in databases. As such, we try to offer an easy-to-read tool for the researcher with details that facilitate the obtention of successful final results.

Reviewer 2 Report

The Authors in the review article have summarized the pathophysiological features of glaucoma, characteristics of all available animal models for various forms of glaucoma disease, and their advantages and disadvantages for research on glaucoma. The review mainly focuses on the pre-clinical information obtained and could be obtained from these models for successful clinical translation. The review is very well written and provides ample information on this subject.

--

The authors should also include another important genetic model of glaucoma i.e. Tyr423His Myoc mice which demonstrate progressive degenerative changes in the peripheral RGC layer and optic nerve (Invest Ophthalmol Vis Sci. 2008 May; 49(5):1932-9, J Neurosci. 2006 Nov 15; 26(46):11903-14).

Author Response

The Authors in the review article have summarized the pathophysiological features of glaucoma, characteristics of all available animal models for various forms of glaucoma disease, and their advantages and disadvantages for research on glaucoma. The review mainly focuses on the pre-clinical information obtained and could be obtained from these models for successful clinical translation. The review is very well written and provides ample information on this subject.

--

-The authors should also include another important genetic model of glaucoma i.e. Tyr423His Myoc mice which demonstrate progressive degenerative changes in the peripheral RGC layer and optic nerve (Invest Ophthalmol Vis Sci. 2008 May; 49(5):1932-9, J Neurosci. 2006 Nov 15; 26(46):11903-14).

We agree with the reviewer's assessment and we have proceeded to add these investigations to the text.

-Moderate English changes required

We appreciate this assessment. The manuscript was given to a native English speaker for corrections.